

# Understanding the response in *Pugionium cornutum* (L.) Gaertn. seedling leaves under drought stress using transcriptome and proteome integrated analysis

Ping Wang*, Zhaoxin Wu*, Guihua Chen and Xiaojing Yu

College of Horticulture and Plant Protection, Inner Mongolia Agricultural University, Hohhot, Inner Mongolia, China
* These authors contributed equally to this work.

## ABSTRACT

**Background:** Drought is one of the crucial constraints limiting horticultural plant's production and development around the world. *Pugionium cornutum* is an annual or biennial xerophyte with strong environmental adaptability and drought resistance; however, the mechanisms with respect to response to drought stress remain largely unclear.

**Methods:** After seedling emergence, the gravimetric method was used to control soil relative water content (SRWC). Drought stress was applied to the six-leaf stage *P. cornutum* seedlings. The soil water content of different drought stress levels (L) was controlled by gravimetric method as follows: control (L1): 70–75% SRWC; moderate drought level (L2): 40–45% SRWC; severe drought level (L3): 30–35% SRWC, and the water was added to different drought stress levels at about 18:00 p.m. every day. The experiment ended when the leaves of *P. cornutum* showed severe wilting (10-leaf stage). Samples were harvested and stored at −80 °C for physiological determination, and transcriptomic and proteomic sequencing.

**Results:** Compared with L1, the leaves of *P. cornutum* seedlings were increasingly wilted after drought treatment; the SRWC of the drought-stress leaves decreased notably while the leaf water potential was rose; the proline, malondialdehyde (MDA) content increased with the continuous drought treatment but peroxidase (POD) activity decreased. Besides, 3,027 differential genes (DGs) and 196 differential proteins (DPs), along with 1,943 DGs and 489 DPs were identified in L2-L1 and L3-L1, respectively. The transcriptome and proteome integrated analysis manifested that only 30 and 70 were commonly regulated both in L2-L1 and L3-L1, respectively. Of which, 24 and 61 DGs or DPs showed the same trend including sHSPs, APX2, GSTU4, CML42, and POD, *etc*. However, most of DGs or DPs were regulated only at the transcriptome or proteome level mainly including genes encoding signal pathway (PYR1, PYLs, SnRK2J, PLC2, CDPK9/16/29, CML9, MAPKs), transcription factors (WRKYs, DREB2A, NAC055, NAC072, MYB and, HB7) and ion channel transporters (ALMT4, NHX1, NHX2 and TPK2). These genes or proteins were involved in multiple signaling pathways and some important metabolism processes, which offers valuable information on drought-responsive genes and proteins for further study in *P. cornutum*.

Corresponding author
Ping Wang, wangping@imau.edu.cn

# INTRODUCTION

As a sort of detrimental abiotic stress for the growth and progression of plants, drought negatively impacts yields and quality (*Wang, Chen & Huang, 2019*). In drought conditions, some plants decrease water loss and enhance water use efficiency by changing the morphological structure of their leaves such as the drooping and wilting leaves (*Luo, Wang & Jin, 2019*), by decreasing plant cell turgor (*Zhang, 2019*). At the same time, reactive oxygen species (ROSs) can be generated by drought conditions, leading to peroxidation of the cell membrane and eventually plant death in extreme environmental cases.

The breeding of crops using molecular genetics to produce high quality has been an effective strategy to promote the sustainable development of agriculture. It is difficult to use model plants including *Arabidopsis thaliana* and *Oryza sativa* in current breeding programs because of their low-stress tolerance, despite the abundance of molecular knowledge derived from them (*Cui et al., 2020*). Numerous xerophytes have evolved to withstand extremely harsh conditions, resulting in forming of multiple protective mechanisms as well as generating genes that resists environmental conditions. In this way, improved drought tolerance in crops can be achieved by understanding the molecular basis of xerophytes' adaptations to drought.

*Pugionium cornutum* (L.) Gaertn, an annual or biennial herb belonged *Brassicaceae*, is a typical xerophyte widely distributed in mobile dunes of the Mu Us Desert, Horqin, and Hulunbuir sandy land in China (*Wang, Wang & Yang, 2017*). Morphological characteristics of *P. cornutum* are pinnate compound leaves, terminal racemes, fruit-owned upper wings, and yellowish and brown seeds (Fig. S1). As a traditional wild vegetable for local people, *P. cornutum* is rich in protein, crude fat, vitamin, and mineral elements. Due to the influence of the original habitat, it is rich in some secondary metabolites especially ascorbic acid and glucosinolates which almost extensively exist in the *Brassicaceae* family. In addition, it has drought tolerance and sand-fixing effect because of its developed root system and strong ability to absorb water. Physiological adaptations of *P. cornutum* to drought have been studied extensively in recent years to understand its drought survival mechanisms (*Hao et al., 2005*; *Li et al., 2016*). In contrast, there have been few reports regarding the molecular mechanisms in *P. cornutum* that contribute to stress tolerance.

Nowadays, integrated multi-omics analyses have been broadly used to illuminate the molecular mechanism of target traits (*Soto-Suárez et al., 2016*). Using a single transcriptome or proteome as a source of information only provides information unilaterally at the transcriptional or translational level (*Ding et al., 2019*; *Yan et al., 2020*; *Wang, Chen & Huang, 2019*). Combined transcriptomic and proteomic analysis is to benefit the study of gene expression regulation at multiple levels, and is a valid method for studying the molecular mechanisms that underlie many biological characteristics (*Chen et al., 2013*; *Du et al., 2021*). Thus, genes and proteins expression profiles and the regulatory pathways of *P. cornutum* leaves under drought stress were studied by the RNA-seq and

Tandem Mass Tag (TMT) techniques. It is possible that this study could provide useful genetic information for the breeding of drought-improved crops.

## MATERIALS AND METHODS

### Plant culture and stress treatment

The collection of plant material in this study complies with relevant institutional, national, and international guidelines and legislation. The fruits of *P. cornutum* from Mu Us Sandland, located in Ordos City in the Inner Mongolia Autonomous Region of China, which do not need specific permits were collected (*Wang, Wang & Yang, 2017*). The collection of plant material in this study complies with relevant institutional, national, and international guidelines and legislation. The seeds were disinfected with 2% NaClO for 10 min, cultured in a petri dish in an incubator at 28 °C in the dark. After approximately 7 days, the germinating seeds sowed into the PVC pot (five seeds/pot) containing 1.5 kg mixture (sand 4: manure 1; v/v) and placed in heliogreenhouse (*Wang, Wang & Yang, 2017*). After seedling emergence, soil relative water content (SRWC) was kept at 70–75% by gravimetric method (*Wang, Wang & Yang, 2017*). The seedlings were thinned at the 4-leaf stage, leaving three healthy and consistent seedlings in each pot. The six-leaf stage seedlings of uniform growth were chosen for the drought treatments. The soil water content of different drought stress levels (L) was controlled by gravimetric method as follows: control (L1): 70–75% SRWC; moderate drought level (L2): 40–45% SRWC; severe drought level (L3): 30–35% SRWC, and the water was added to different drought stress levels at about 18:00 p.m. every day. The experiment ended when the leaves showed severe wilting (10-leaf stage) under severe drought stress. Samples were harvested at 7:30 a.m. of the next day, and the leaves and roots were cut into pieces, mixed, respectively, then immediately frozen and stored at −80 °C for subsequent sequencing.

### Physiological determination

To understand the changes in physiological parameters under control and drought stress in *P. cornutum* seedings, five indicators including relative water content, water potential, malondialdehyde (MDA), proline content, and peroxidase (POD) activity were measured in *P. cornutum* leaves at the six-leaf stage (triple replicates) (*Zhang et al., 2016*). Relative water content was determined by the oven drying method (*Zou, 2000*). Water potential was measured by small fluid flow (*Bai et al., 2012*). The content of MDA and proline were determined by the thiobarbituric acid method and acid ninhydrin colorimetry, respectively (*Zhao & Cang, 2016*). The POD activity was determined by ultraviolet spectrophotometry (*Li, 2000*).

### RNA isolation

As directed by the manufacturer, total RNA was extracted from the *P. cornutum* seedling leaves (L1, L2, and L3, three replicates each) using RNAprep Pure Plant Plus Kit (TIANGEN, Beijing, China). RNA samples were determined for degradation and impurities by 1% agarose electrophoresis. RNA purity was checked using the Kaiao K5500® spectrophotometer (Kaiao, Beijing, China). RNA integrity and concentration

were assessed using the RNA Nano 6000 Assay Kit of the Bioanalyzer 2100 system (Agilent Technologies, CA, USA). A total of nine high-quality RNA samples were acquired and used for RNA-seq and qRT-PCR analysis.

### cDNA library construction and *de novo* transcriptome assembly

Three different drought-level RNA samples (each sample with three biological repeats) were used for cDNA library construction. Illumina HiSeq X Ten was used to sequence both the 5′ and 3′ ends, generating over 300 million paired reads (*Wang, Wang & Yang, 2017*). Obtaining clean reads used for transcriptomic analysis. All the following analyses relied on high-quality clean data. Trinity software was used for *de novo* assembly which is especially robust without a reference genome. The statistical power of this experimental design, calculated in *Power analysis* (https://rodrigo-arcoverde.shinyapps.io/rnaseq_power_calc/) is 0.9458904.

### Protein preparation

SDT buffer (4% SDS, 100 mM Tris-HCl, 1 mM DTT, pH7.6) was used for tissue of *P. cornutum* leaves lysis and protein extraction. After agitating the tissue with a homogenizer and boiling it for 5 min, the samples were ultrasonically disrupted and boiled for another 5 min. By centrifuging at 28,500 g for 15 min, undissolved cellular debris was removed. A BCA Protein Assay Kit (Protein Assay Kit; Bio-Rad, Hercules, CA, USA) was used to quantify the supernatant (*Wisniewski et al., 2009*).

### Protein sequencing and bioinformatics analysis

*P. cornutum* leaves (200 μg) were used for protein digestion according to *Wisniewski et al. (2009)*. A TMT reagent was used to label the peptides based on the manufacturer's instructions (*Stryiński et al., 2019*). Mass Spectrometry (LC-MS) analysis was performed using a Q Exactive mass spectrometer combined with an Easy nLC (Thermo Fisher Scientific, Waltham, MA, USA). MaxQuant software (version 1.6.0.16) was used to import raw LC-MS/MS files. Perseus software, Microsoft Excel, and R software were used for analyzing the data. FoldChange >1.20 or <0.83 and $P < 0.05$ were regarded as the screening criteria for DPs. Hierarchical clustering was performed according to protein expression level. A Fisher exact test was considered as examining GO and KEGG enrichment. FDR correction for multiple testing was also carried out (*Ashburner et al., 2000*).

### Quantitative polymerase chain reaction (qPCR) analysis

RNA sample was reversely transcribed to cDNA according to instruction of Universal RiboClone® cDNA Synthesis System (Promega, Beijing, China). qPCR was conducted using TB Green Premix Ex Taq II (TliRNaseH Plus) (RR420Q TaKaRa Biotechnology, Beijing, China) on an ABI7500 system (Thermo Fisher, Singapore, Asia). The reaction procedure was completed with the following protocol: 95 °C 3 min, 45 cycles of 5 s at 95 °C, 30 s at 60 °C and 71 cycles of 15 s at 60 °C. All samples were tested with three technical replicates and three independent biological replicates. The relative expression level was calculated while using the $2^{-\Delta\Delta CT}$ method (*Livak & Schmittgen, 2001*). According

Peer J

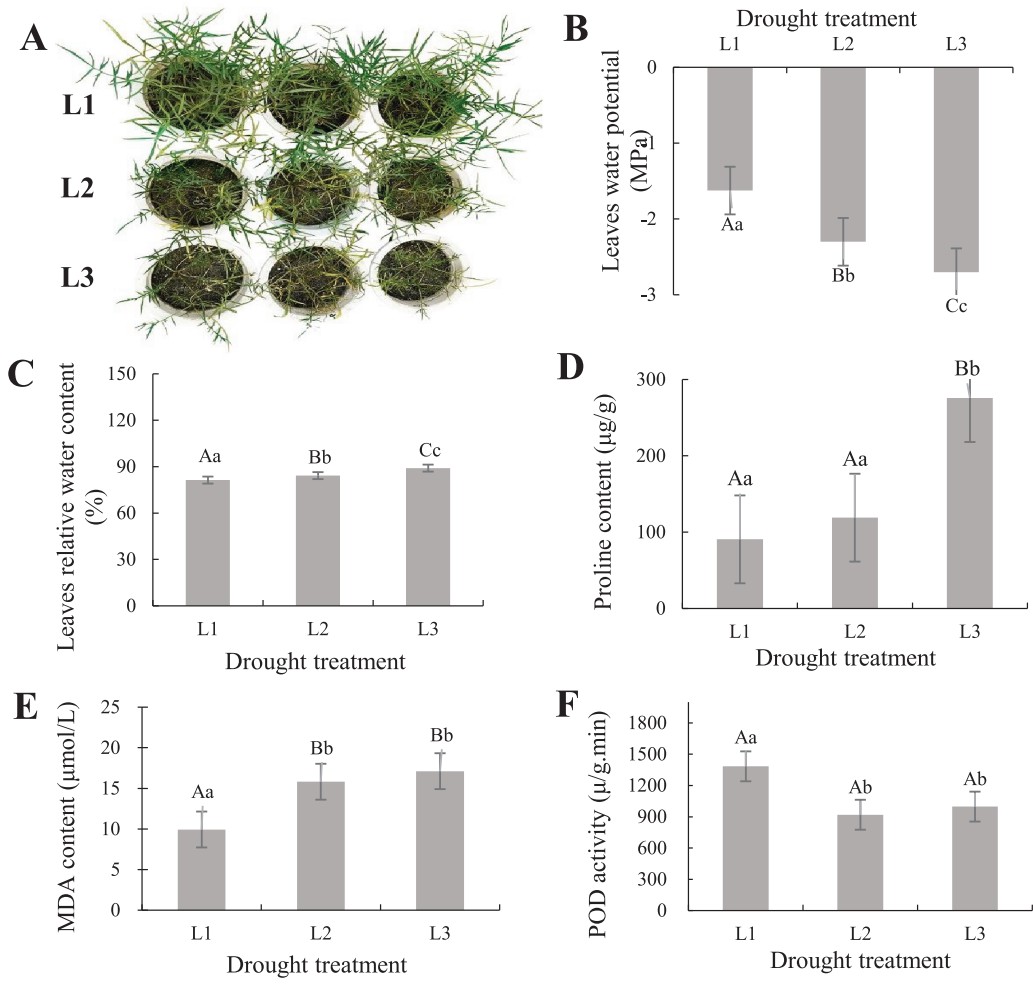

**Figure 1 (A–F) Phenotypic status and determination of physiological indicators of *P. cornutum* under drought treatment.**

to a previous study in *P. cornutum*, the beta-actin gene was regarded as a reference gene (*Wang, Wang & Yang, 2017*).

## Data analysis

Physiological data analysis was conducted ANOVA using SPSS 19.0 software (SPSS Inc., Chicago, IL, USA) and Microsoft 2016 (Microsoft Corporation, Redmond, CA, USA). Duncan's multiple range test was used to detect a difference between means at a significance level of $P < 0.05$.

## RESULTS

### Physiological responses of *P. cornutum* in drought stress

To understand the phenotypic changes and physiological indicators of the leaves of *P. cornutum*, we treated six-leaf-stage seedlings with various drought stress. The result showed *P. cornutum* leaves were increasingly wilted under progressive drought (Fig. 1A). Fresh leaf samples were used to determine SRWC, proline, MDA content, and POD

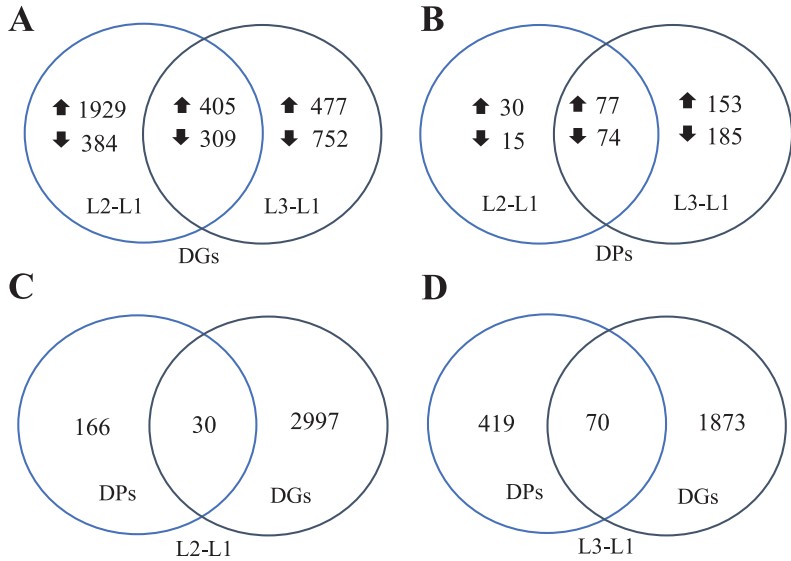

**Figure 2** Venn diagrams of DGs (A), DPs (B), comparing DPs and DGs in L2-L1 (C) and L3-L1 (D) in *Pugionium cornutum* leaves after drought treatment.

activity. The leaves water potential was rose (Fig. 1B), while the SRWC of the drought-stress leaves decreased notably under L3 and L2 compared with L1 treatment (Fig. 1C). The proline, MDA content increased with the continuous drought treatment but POD activity decreased (Figs. 1D–1F). The plant physiological results were consistent with the observed phenotypic changes, which indicated *P. cornutum* simultaneous responded to drought through phenotypic and physiological changes. Next, *P. cornutum* leaves were harvested for use in transcriptome and proteome analyses under drought stress.

## Transcriptome profiling of *P. cornutum* seedling leaves in drought stress

To clearly understand the transcriptomic data and identify DGs in *P. cornutum* during drought stress, we performed RNA-seq on libraries prepared from the leaf tissue of control and drought-stressed *P. cornutum*. Raw data were yielded from L1, L2 and L3 sample, which were 168302246, 175410328, and 167416966, respectively. After filtering the raw data, 125134518, 136036228, and 129590740 clean reads were acquired. Average length of unigenes after transcript assembly was 749 bp. DGs were defined with FDR <0.05 and $\log_2|foldchange| >1$. In total, 3,027 (2,334 upregulated and 693 downregulated) and 1,943 (882 upregulated and 1,061 downregulated) DGs were identified as responsive to drought stress in L2-L1 and L3-L1, respectively (Fig. 2A). Among up-regulated DGs, 1,929 DGs were exclusively in L2-L1, 477 DGs were exclusively in L3-L1 and 405 DGs were both in L2-L1 and L3-L1. Bsides, among down-regulated DGs, 384 DGs were exclusively in L2-L1, 752 DGs were exclusively in L3-L1, and 309 DGs were both in L2-L1 and L3-L1 (Table S1). The identified DGs were enriched in KEGG under different drought stress (Fig. 3). Of these, common pathways included protein processing in endoplasmic reticulum, phenylalanine metabolism, phenylpropanoid biosynthesis, starch and sucrose metabolism, glucosinolate biosynthesis and plant hormone signal transduction. GO analysis assisted in

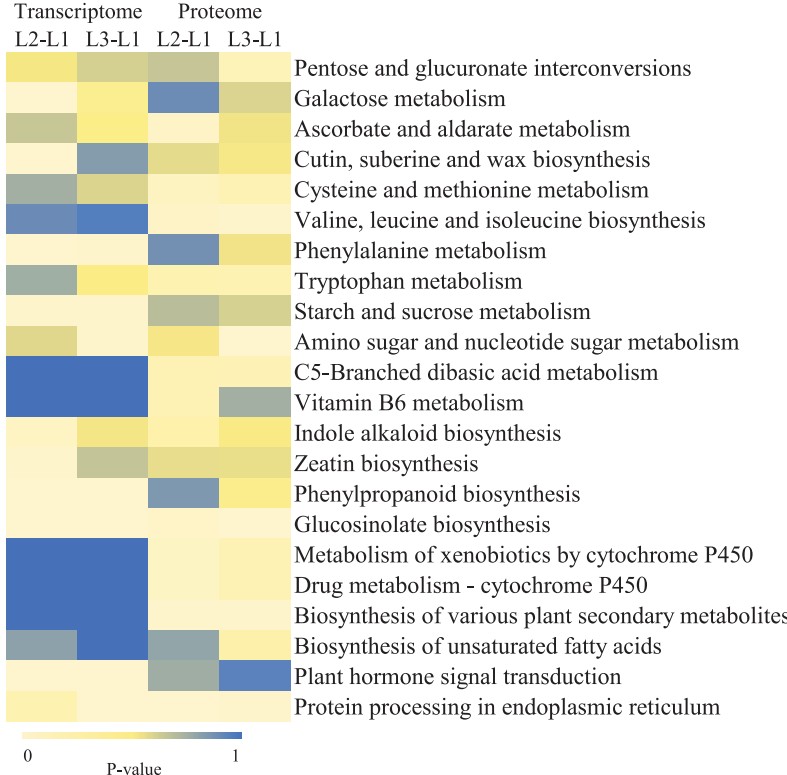

Transcriptome   Proteome
L2-L1  L3-L1  L2-L1  L3-L1

Pentose and glucuronate interconversions
Galactose metabolism
Ascorbate and aldarate metabolism
Cutin, suberine and wax biosynthesis
Cysteine and methionine metabolism
Valine, leucine and isoleucine biosynthesis
Phenylalanine metabolism
Tryptophan metabolism
Starch and sucrose metabolism
Amino sugar and nucleotide sugar metabolism
C5-Branched dibasic acid metabolism
Vitamin B6 metabolism
Indole alkaloid biosynthesis
Zeatin biosynthesis
Phenylpropanoid biosynthesis
Glucosinolate biosynthesis
Metabolism of xenobiotics by cytochrome P450
Drug metabolism - cytochrome P450
Biosynthesis of various plant secondary metabolites
Biosynthesis of unsaturated fatty acids
Plant hormone signal transduction
Protein processing in endoplasmic reticulum

0        P-value        1

**Figure 3 KEGG pathways enrichment analysis of DGs and DPs in L2-L1 and L3-1 after drought sress.**

further elaborating the functions of these DGs. In L2-L1, most annotated DGs were enriched in response to abiotic stimulus (GO:0009628), hormone-mediated signaling pathway (GO:0009755), and shoot system development (GO:0048367) in the biological process, transcription factor activity (GO:0003700), nucleic acid binding transcription factor activity (GO:0001071) and protein serine/threonine kinase activity (GO:0004674) in molecular function and symplast (GO:0055044) and cell-cell junction (GO:0005911) in the cellular component (Fig. 4A). In L3-L1, annotated DGs were enriched in response to acid chemical (GO:0001101), response to hormone (GO:0009725), hormone-mediated signaling pathway (GO:0009755) and response to oxygen-containing compound (GO: 1901700) in the biological process and transcription factor activity (GO:0003700) in the molecular function (Fig. 4B).

## Proteome profiling of *P. cornutum* seedling leaves in drought stress

Meanwhile, a proteomics analysis was performed for the above leaves of *P. cornutum* to complement the transcriptome study using the TMT approach. Approximately 335,043 total spectra were generated from leaves. After eliminating low-scoring spectra, 13,162 unique peptides and 4,588 protein groups were detected. Proteins with a ratio fold change of >1.2 or <0.83 and a *p*-value < 0.05 were identified as DPs. In total, 196 (107 upregulated and 89 downregulated) and 489 (230 upregulated, 259 downregulated) DPs were identified in L2-L1 and L3-L1, respectively (Fig. 2B). Among up-regulated DPs, 30 DPs were

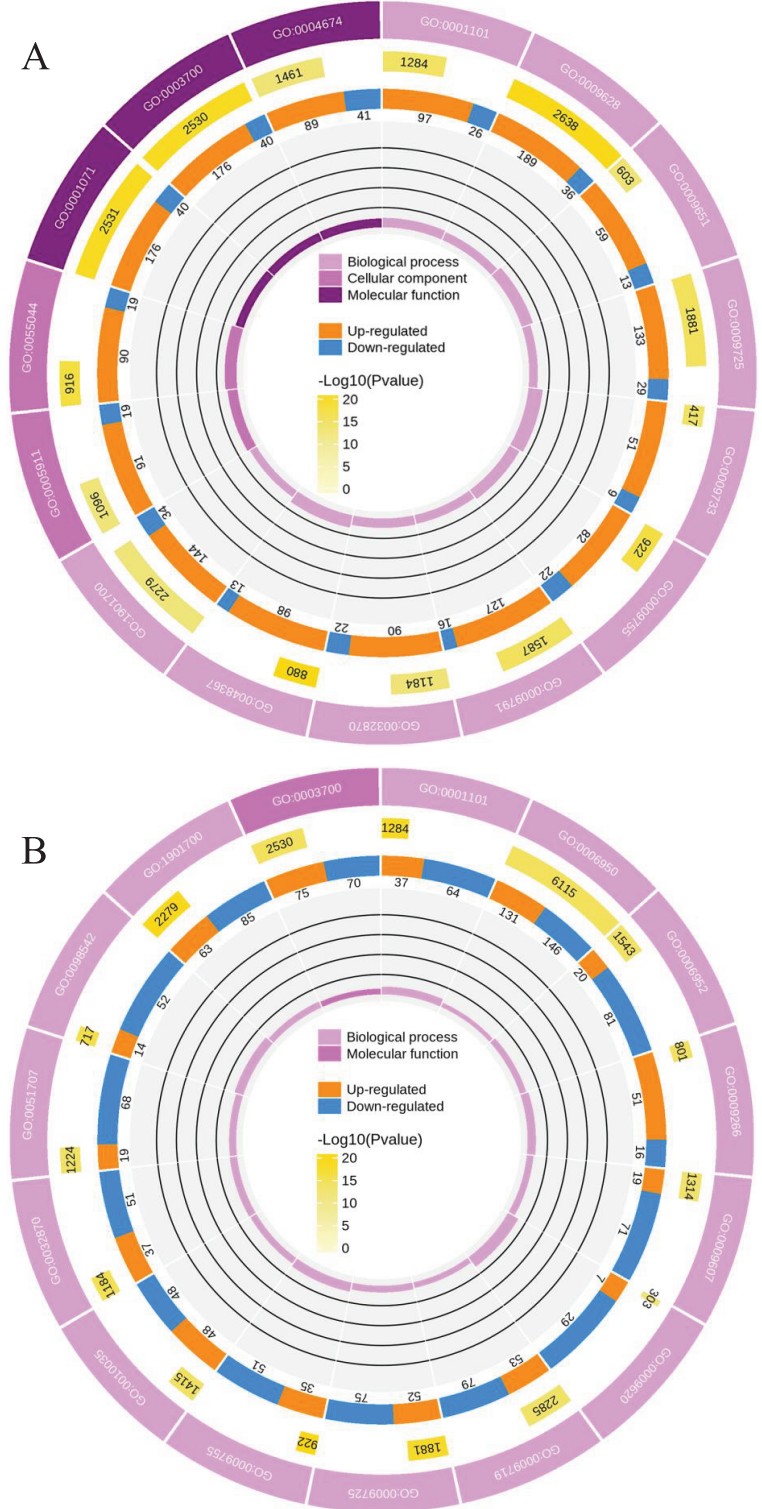

**Figure 4 Functional classification of DGs based on GO categorization in L2-L1 (A), L3-L1 (B).**

exclusively in L2-L1, 153 DPs were exclusively in L3-L1, 77 DPs were both in L2-L1 and L3-L1. Among down-regulated DPs, 15 DPs were exclusively in L2-L1, 185 DPs were exclusively in L3-L1, and 74 DPs were both in L2-L1 and L3-L1 (Table S2). Moreover, these DPs were significantly enriched in cysteine and methionine metabolism, valine, leucine, and isoleucine biosynthesis, tryptophan metabolism, C5-Branched dibasic acid metabolism, glucosinolate biosynthesis, metabolism of xenobiotics by cytochrome P450, biosynthesis of various plant secondary metabolites and protein processing in endoplasmic reticulum (Fig. 3). These DPs were assigned GO terms and they were enriched in response to reactive oxygen species (GO:0000302), response to stress (GO:0006950) and response to heat (GO:0009408) in biological processes, ferroxidase activity (GO:0004322) in molecular functions in L2-L1 (Fig. 5A). In L3-L1, annotated DGs were enriched in response to stress (GO:0006950), response to temperature stimulus (GO:0009266), response to heat (GO: 0009408) and response to abiotic stimulus (GO:0009628) in biological processes and hydrolase activity (GO:0004553) and hydrolase activity (GO:0016798) in molecular functions (Fig. 5B).

## Validation of transcriptomic and proteome data

To validate the proteomic data, twelve randomly proteins were selected for qPCR analysis, all of which were associated with drought stress. The above 12 proteins were homologously compared to our transcriptome database, of which seven mRNA sequences were found, and qPCR primers were designed (Table S3). These proteins embodied different expression levels (Fig. 6). Only one protein was inconsistent with the transcriptomic and proteomic data, the other six were the same expression trend, which confirmed the reliability of the transcriptomic and proteomic data.

## Transcriptomic and proteomic correlation analysis

Transcriptome and proteome profiles subject to drought stress were analyzed. Next, correlation analysis was performed to assess the data relation between RNA-seq and TMT and the correlation coefficient in L2-L1 ($R^2 = 0.657$), L3-L1 ($R^2 = 0.789$) indicated DP and their corresponding mRNA have certain biological relevance in response to drought in *P. cornutum* seedling leaves (Fig. 7).

## Integrated transcriptomic and proteomic analysis in *P. cornutum* seedling leaves

In summary, 3,027 and 1,943 DGs were identified in L2-L1 and L3-L1, respectively. Meanwhile, 196 and 489 DPs were identified in L2-L1and L3-L1, respectively. More DGs were identified in L2-L1 than DPs, which showed *P. cornutum* reacted quickly to the changes in gene expression in moderate drought stress. In addition, only 30 and 70 were commonly regulated both at the transcriptome and proteome levels (Fig. 2C and 2D). Of which, 24 and 61 DGs or DPs showed the same trend, indicating that these proteins were mainly controlled by corresponding genes changes. It is worth noting that six and nine DGs or DPs belonging to heat shock proteins (HSPs) were dramatically induced in L2-L1 and L3-L1, which exhibited HSPs act as the positive role response to drought in
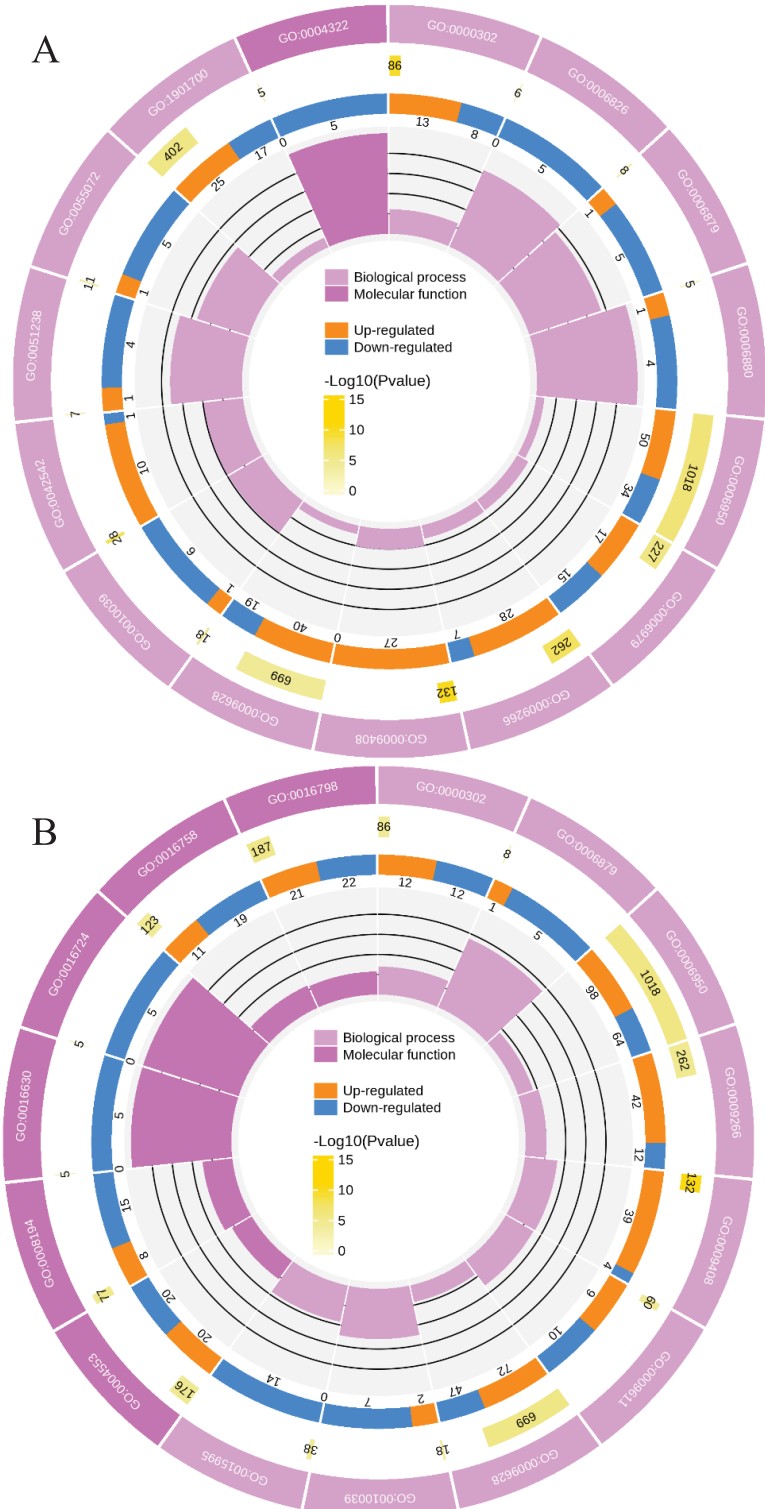

**Figure 5 Functional classification of DPs based on GO categorization in L2-L1 (A), L3-L1 (B).**

 

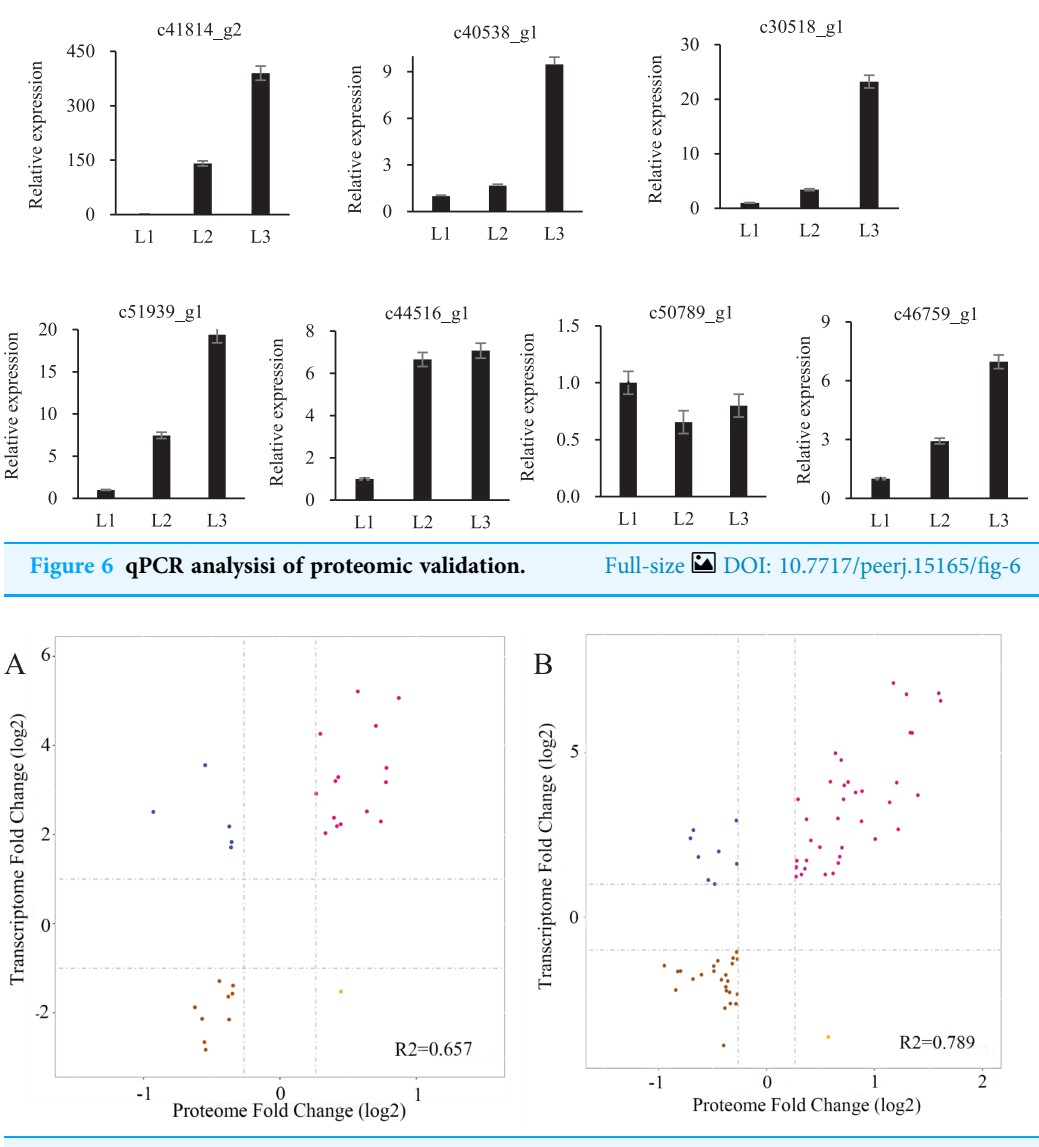

**Figure 6 qPCR analysisi of proteomic validation.**

**Figure 7 Correlation between the expression of DGs and DPs in L2-L1 (A), L3-L1 (B).** Purple and brown dots represent significant up-regulated and down-regulated DPs, respectively and blue dots represent nonsignificant DPs.

*P. cornutum* seedling leaves. Similarly, some DGs or DPs involved in ROS scavenging, for instance, ascorbate peroxidase 2 (APX2), glutathione S-transferase U4 (GSTU4), and galactinol synthase (GolS1), were greatly up-regulated. Beta-galactosidase 1 and 4 (BGAL1 and BGAL4) involved in cell wall polysaccharide metabolism were significantly induced (Table S3). On the contrary, the calcium-binding protein (CML42) involved in calcium signaling was suppressed by drought stress. Consistently, glucosinolate biosynthesis related genes or proteins, including cytochrome P450 77A4 (CYP77A4), cytochrome P450 83B1 (CYP83B1), and cytosolic sulfotransferase 16 (SOT16) were severely depressed as well (Fig. 8). However, a small number of genes showed opposite transcriptional and protein expression trends in L2-L1 and L3-L1, such as thioredoxin H5 (TRXH5) related to redox metabolism, D-alanine–D-alanine ligase (DDL)| relevant to amino and nucleotide

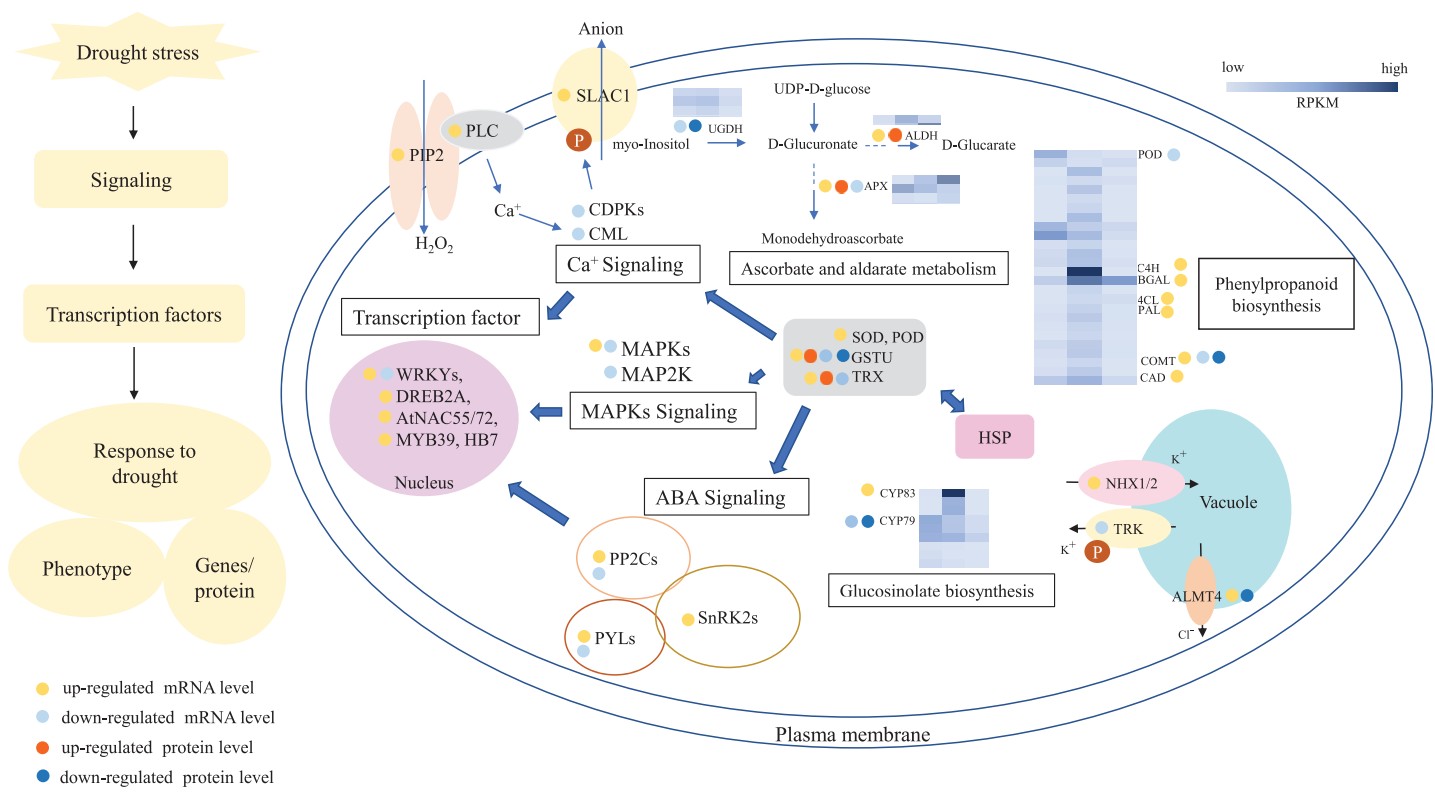

**Figure 8 A outline diagram of transcriptomic and proteomic changes in *P. cornutum* seedling leaves under drought stress.**

sugar metabolism, 1-aminocyclopropane-1-carboxylate oxidase 5 (ACCO5) related to ethylene biosynthetic process.

However, most DGs or DPs were regulated only at the transcriptome or proteome level. There were 1,929 upregulated, 384 downregulated and 477 upregulated, 752 downregulated DGs in L2-L1 and L3-L1, respectively, but the corresponding proteins were not differentially expressed. Accordingly, there were 30 upregulated, 15 downregulated and 153 upregulated, 185 downregulated DPs in L2-L1 and L3-L1, respectively, the corresponding genes were not differentially expressed (Tables S5 and S6).

According to the analysis of DGs or DPs, drought greatly induced or inhibited the expression of genes or proteins involved in ascorbate and aldarate metabolism, such as two up-regulated and one down-regulated APX, two down-regulated UDP glucose 6-dehydrogenase (UGDH) and two up-regulated aldehyde dehydrogenase (ALDH). And some genes or proteins related to antioxidant enzymes and antioxidants were differentially expressed and most of them were up-regulated, such as SOD, POD, GST, PAL, and thioredoxins (TRX). Meanwhile, drought also notably induced or suppressed the expression of genes associated with ABA, Ca+ and MAPKs signaling pathways, such as PYR (PYR1), PYL (PYL4, PYL6, PYL7, PYL8, and PYL11), SnRK2, slow anion channel 1 homolog 2 (SLAC), PIP1-2, PLC2, CDPK9/16/29, CML9, MAPK7 MAPK4 and MAP2K6 (Fig. 8). Furthemore, transcription factor (WRKYs, DREB2A, AtNAC55/72 and MYB) and

channel transport proteins (NHX1/2, TPX, and ALMT4) were differentially expressed in *P. cornutum* under drought stress.

## DISCUSSION

Drought dramatically restricts plants growth and development, they adjust themselves to the adverse circumstance by morphological, physiological, and biochemical changes (*Wang, Chen & Huang, 2019*; *Sun et al., 2019*; *Wen et al., 2020*). According to phenotypic observation, the leaves of *P. cornutum* were increasingly wilted under progressive drought stress (Fig. 1A). Phenotypic changes in plants can be observed under drought stress, however, changes in physiological and molecular events need to be further investigated (*Zheng et al., 2018*). Under drought stress, the relative water content of leaves of *P. cornutum* continuously decreased (*Zhang et al., 2015*) and proline content increased (*Zheng, 2018*). In this study, the SRWC decreased notably under L3 and L2 compared with L1 treatment, while the leaf water potential was rose. The proline, MDA content increased with the continuous drought treatment but POD activity decreased (Fig. 1D), which indicating that *P. cornutum* responds to drought stress through phenotypic and physiological changes.

Plant adaptation to drought is a complicated biological process concerning the cross-regulation of multiple signaling pathways. With the fast growth of sequencing technology, huge amounts of genes related to drought have been discovered. They are usually divided into functional and regulatory genes. The former encodes crucial enzymes as well as proteins related to metabolisms, such as ion transporters ($K^+$ and $Cl^-$ transporters) and HSPs, which directly protect cells and tissue from stress (*Andrés et al., 2014*; *Eisenach et al., 2017*; *Isner et al., 2018*; *Mogk, Ruger-Herreros & Bukau, 2019*). The latter encodes a variety of regulatory proteins, containing transcription factors (DREB, WRKY, *etc.*), protein phosphatases and protein kinases (MAPK), that adjust gene expression and signal network in the adverse situation (*Tran et al., 2004*; *Qin et al., 2007*; *Culter et al., 2010*; *Wang et al., 2020a*). Importantly, the discovery and characterization of these genes and proteins have offered viable genetic strategies for improving the stress resistance of horticultural plants. To understand the molecular mechanism in *P. cornutum* under drought stress, RNA-Seq and TMT methods were conducted. All in all, 6,639 DGs and 795 DPs were quantified. The number of up-regulated DGs was higher than that of down-regulated DGs in L2-L1, indicating the gene expression of *P. cornutum* was active after moderate drought stress. But in L3-L1, the number of down-regulated DGs was slightly more than that of up-regulated DGs, this might be due to the drought stress being too long, resulting in partial gene down-regulated expression. Interestingly, 109 were commonly regulated at both the transcriptomic and proteomic levels, mainly including HSP and antioxidant enzymes (POD, APX, GST, PAL, and TRX). However, most DGs or DPs were regulated only at the mRNA or protein level, which indicated transcriptional and translational realignment may be the major regulatory mechanism of *P. cornutum* seedling leaves response to drought.

## HSPs involved in drought response of *P. cornutum*

Numerous researchers have reported HSPs participated in many abiotic stresses, such as drought stress (*Ding et al., 2019*; *Zhang et al., 2020b*). According to molecular weights, plant HSPs can be divided into sHSPs (15–42 kDa), HSP70 and HSP90, *etc.* (*Waters, Lee & Vierling, 1996*). sHSPs be known as molecular chaperones and it can prevent protein misfolding (*Mogk, Ruger-Herreros & Bukau, 2019*). It was shown that 76% HSPs were significantly increased in cassava leaves after PEG treatment (*Ding et al., 2019*). Some plants were also correlated with the abundance of HSPs such as potatoes (*Zhang et al., 2020b*) and *Sorbus pohuashanensis* (*Zhang et al., 2020a*). Interestingly, there were six up-regulated HSPs in both L2-L1 and L3-L1 and they all belonged to sHSPs, such as c30518_g1, c34946_g2, c40538_g1, c41814_g2, c42546_g1 and c49280_g1 (Table S3), demonstrating that *P. cornutum* may prevent protein misfolding by increasing the expression of HSPs under drought stress. The specific function of these HSPs should be elucidated in further study.

## Antioxidative enzymes involved in drought response of *P. cornutum*

In the leaves of *P. cornutum*, ROSgradually accumulated to adapt drought threat (*Pang et al., 2013*). With intensified drought stress, antioxidant enzymes and antioxidants including SOD, POD, CAT, PAL, APX, GST, and TRX were synthesized by the antioxidant metabolic pathway (*Akcay et al., 2010*; *Basu et al., 2010*). The expressions of antioxidant enzymes were profoundly increased under drought periods in transgenic sweet potato (*Wang et al., 2017*), rice (*Jain, Ghanashyam & Bhattacharjee, 2010*), Melinjo (*Arum et al., 2018*) and Sorghum varieties (*Goche et al., 2020*). In our research, SOD (c43152_g1, c37784_g1), POD (c48398_g1, c40498_g1, c45037_g1, c52744_g2, c52385_g3, c49375_g1, c45077_g1, c51800_g3, c34498_g1, c40491_g1, c46883_g1, c44183_g2, c47873_g1, c52385_g1), GSTU4 (c41814_g1), PAL (c40029_g1, c45828_g1, c46053_g2, c45828_g2, c40029_g1, c46053_g1), and TRXH5(c39489_g1, c39489_g1, c37172_g1) genes were differentially expressed and most of them were up-regulated (Fig. 8), which suggested these up-regulated genes may regulate their enzymes activity and enhance the ability of scavenge ROS to protect *P. cornutum* seeding leaves from drought damage. Moreover, some genes or proteins involved in ascorbate and aldarate metabolism were differentially expressed under drought, such as three APX (c46714_g1, c48058_g1, c41895_g1), two down-regulated UGDH (c51033_g1, c47548_g1) and 1 up-regulated ALDH (c48759_g1) (Tables S3 and S5 and Fig. 8) and their expression levels may be responsible for the high ascorbic acid content in *P. cornutum*. The findings of these genes may provide a theoretical basis for further study on the drought response of *P. cornutum*.

Furthermore, previous studies have shown that complex interactions exist between HSPs and ROSs scavenging enzyme responses in plants (*Ding et al., 2019*; *Driedonks et al., 2015*). Heat shock factors (HSFs) stimulate the expression of HSP and also affect ROS scavenger gene expression under heat stress (*Driedonks et al., 2015*). Interactions between HSPs and ROSs scavenging enzymes in *P. cornutum* under drought stress will be increasingly attached importance to future work.

## Genes related to the signaling pathway involved in the drought response of *P. cornutum*

Generally, the plant quickly delivers stress signals and make appropriate regulations when they encountered adverse environmental conditions (*Gong et al., 2020*). There are three main signal pathways involved in plant drought stress including ABA pathway, $Ca^{2+}$ pathway, and MAPKs pathway (*Wang et al., 2020a*). Drought stress induced the accumulation of ABA, ABA receptor (PYR/PYL/RCAR) senses ABA signal and binds to it, it forms ABA-PYR/PYL/RCAR complex that can inhibit PP2Cs activity and release SnRK2s (*Culter et al., 2010*). Subsequently, SnRK2s phosphorylate corresponding transcription factors, including WRKYs, and DREB2A that cause the expression of ABA and drought genes, thereby enhancing the drought resistance of plants (*Wang et al., 2020b*). ZmDREB2A, AtNAC019/055/072 and HB7/12 were strongly inducible by drought and activated corresponding gene expression (*Tran et al., 2004*; *Qin et al., 2007*).

In Strikingly, many genes (PYR1, PYL4, PYL6, PYL7, PYL8, PYL11, and SnRK2J) and transcription factors (WRKYs, DREB2A, NAC055, NAC072, MYB, and HB7) referred to ABA signaling were also dramatically induced in our study (Table S5 and Fig. 8), which show these genes and transcription factors might be the key response to drought stress through ABA signaling in *P. cornutum* seedling leaves. In addition, we have also found that slow anion channel 1 homolog 2 (SLAC, c49362_g1) phosphorylated by SnRK2s was also induced which may be the reason for stomatal closure and improved drought resistance in *P. cornutum* seedling leaves (Fig. 8).

Under drought stress, extracellular $Ca^{2+}$ binds to calcium receptors on the membrane of plants, activating phospholipase C (PLC) and hydrolyzing with phosphatidylinositol-4,5-bisphosphate (PIP2), resulting in increased intracellular $Ca^{2+}$ levels and then it is further transmitted by calcium receptors CML, CBLs and CDPKs, and regulated the expression of related genes by phosphorylation of downstream target proteins (*Wang et al., 2020a*). It's worth noting that several genes involved in the $Ca^{2+}$ signaling pathway were regulated such as PIP1-2 (c35594_g1), PLC2 (c47765_g1, c47765_g5), CDPK9/16/29 (c38659_g1, c38659_g2, c44700_g1), and CML9 (c45168_g1), indicating that the expression of these genes of the $Ca^{2+}$ signaling pathway was affected by drought stress in *P. cornutum* seedling leaves.

In response to drought stress, MAP kinase kinase kinase kinase (MAP4K) or receptor protein on the plasma membrane is activated to MAPK by cascade phosphorylation and MAPK phosphorylates downstream target proteins to regulate the expression of the related genes, thus enhancing drought resistance. In this work, MAPK7 (c45059_g2, c45059_g1) were greatly up-regulated and MAPK4 (c51134_g1) and MAP2K6 (c35626_g1) were down-regulated. These genes exhibited different expression level, one part was positive regulation and the other was negative regulation, which may be caused by cross talks of multiple signaling pathways and metabolic processes. Besides, $K^+$ and $Cl^-$ homeostasis in the vacuoles is critical for stomatal movement. ALMT4, $Cl^-$ channel transporter, is likely activated by MAPKs and mediates $Cl^-$ efflux from the vacuoles in drought stress. The atalmt4 mutants were more sensitive to drought stress than the wild (*Eisenach et al., 2017*).

Consistently, 2 ALMT4 (c37684_g1, c36137_g1) genes were significantly induced in this work, suggesting that the drought resistance of *P. cornutum* may also be enhanced by Cl⁻ efflux under drought stress. NHX mediates $K^+$ uptake into and TPK1 transports $K^+$ out of the vacuoles by the proton gradient (*Andrés et al., 2014*; *Isner et al., 2018*). Amazingly, NHX1 and NHX2 (c40067_g1, c52761_g3) were dramatically up-regulated and TPK2 (c31434_g1) was severely depressed in this work that demonstrated the accumulation of $K^+$ may promote the stomatal closure and reduced water loss of *P. cornutum* seedling leaves under drought circumstances.

Together, these findings implied that ABA, $Ca^+$, and MAPK signaling pathways may be involved in the response of *P. cornutum* seedlings leaves to drought stress, and the three signaling pathways may have an effect on the $K^+$ and Cl⁻ homeostasis. Further studies should focus on how the genes involved in the three signaling pathways regulate the metabolic pathways of *P. cornutum* seedlings leaves to improve their ability to resist drought and identify the functions of specific genes corresponding to specific signaling pathways.

## CONCLUSIONS

This study provides the first integrated analysis of transcriptomic or proteomic in *P. cornutum* seedlings leaves under drought stress. Our study showed that facilitating the expression of sHSPs and antioxidant enzymes (APX2, GSTU4, CML42, and POD, *etc.*) are likely to confer drought tolerance to *P. cornutum*. Meanwhile, possible transcription factor involved in drought stress (WRKYs, DREB2A, NAC055, NAC072, MYB, and HB7), some genes encoding signal pathways (PYR1, PYLs, SnRK2J, PLC2, CDPK9/16/29, CML9, MAPKs) and ion channel transporters (ALMT4, NHX1, NHX2 and TPK2) were identified. These results promote study on the molecular mechanism of drought tolerance in *P. cornutum*. In the future, we will focus on the specific functions of the above genes and proteins and how they regulate the responses of *P. cornutum* seedling leaves to drought.

## ACKNOWLEDGEMENTS

We highly acknowledge the technical support of the company of Anoroad and Shanghai Applied Protein Technology.

### Funding
This work was supported by grants from the National Natural Science Foundation of China (Grant Nos. 31860553 and 31260475). The funders had no role in study design, data collection and analysis, decision to publish, or preparation of the manuscript.

### Grant Disclosures
The following grant information was disclosed by the authors:
National Natural Science Foundation of China: 31860553 and 31260475.

## Competing Interests

The authors declare that they have no competing interests.

## Author Contributions

- Ping Wang conceived and designed the experiments, prepared figures and/or tables, authored or reviewed drafts of the article, and approved the final draft.
- Zhaoxin Wu performed the experiments, analyzed the data, prepared figures and/or tables, authored or reviewed drafts of the article, and approved the final draft.
- Guihua Chen performed the experiments, prepared figures and/or tables, and approved the final draft.
- Xiaojing Yu performed the experiments, prepared figures and/or tables, and approved the final draft.

## Data Availability

The RNA-Seq are available in the Short Read Archive (SRA) of the NCBI: PRJNA730048.

The proteomics data are available at the ProteomeXchange Consortium *via* the iProX partner repository: IPX00031710.

## Supplemental Information

Supplemental information for this article can be found online at http://dx.doi.org/10.7717/peerj.15165#supplemental-information.

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
