# Peer review of "Understanding the response in Pugionium cornutum (L.) Gaertn. seedling leaves under drought stress using transcriptome and proteome integrated analysis"

_PeerJ, doi:10.7717/peerj.15165_

## Round 0.1 · original submission · Major Revisions

We have received 3 critical reviews demanding revision. The comments are very detailed. It will help to improve the text. Please check English, ask for help from a proficient speaker for better understanding, and to reach international readers.

·

Basic reporting

In the manuscript entitled " Understanding the response in Pugionium cornutum (L.) Gaertn. seedling leaves under drought stress using transcriptome and proteome integrated analysis”, Wang et al. have done on transcriptomics and proteomics analysis to unravel drought-responsive genes and proteins in Pugionum. The manuscript describes the response of a xerophytic species to drought stress which is expected to shed more light on underlying regulatory mechanism. In general, it is a good piece of information however special care should be taken to address the issue of English language to enable the contents to be clearly understandable to international audience. Many abbreviations in the manuscript are not written in their expanded form. Few suggestions as mentioned may be seen by authors for incorporation.

Experimental design

Reference for the methodology used for the measurement of relative water content (RWC), water potential, malondialdehyde (MDA), proline content, catalase activity (CAT) and other experimental procedures should be cited.

Figure 1 needs to be re-organized for better representation.
Quality of Figure 2, 3, 4, 5 and 6 needs to be improved.

Validity of the findings

In the result section authors have shown that transcript level for L2-L1 treatment showed up-regulation for maximum number of DGs (3027) compared to L3-L1 (1943), contrarily for proteome profiling L3-L1 showed higher number of DPs (489) compared to L2-L1 (196). Why the higher transcript abundance did not correlate with the protein turnover for respective treatment?

Discussion part needs to be re-written as it highlights most of the major signaling pathways (ROS, MAPK, ABA, WRKY, DREB, CDPKs, etc.) involved in abiotic stress tolerance which makes it more of a generalized outcome rather than the salient finding of the present study. The author could sub categorize discussion as per the findings while establishing a correlation between them with respect to current study if any. For e.g.: Sub categorizing as in: Physiological response, role of metabolic synthetic pathways, role of stress responsive pathways/genes etc.

Additional comments

The English language and typographical errors need to be taken care of.

·

Basic reporting

In the manuscript entitled " Understanding the response in Pugionium cornutum (L.) Gaertn. seedling leaves under drought stress using transcriptome and proteome integrated analysis”, authors have conducted transcriptome and proteome analysis in Pugionum, a xerophytic plant species in response to drought stress. The present study will give insights in elucidation of molecular mechanism and the regulatory mechanism underlying drought tolerance in other crops. Several genes and mechanism have been proposed in the present study, however the authors need to restructure the manuscript in terms of english language and sentence framing to have a wider reach and better understanding of readers.
Few suggestions have been mentioned herein that needs to be addressed by the authors:
1.Full form of many abbreviations throughout the manuscript has not been mentioned. It is suggested that author should mention the full form of used abbreviations for better understanding of the readers. For e.g.: in Line no. 41 water holding capacity for WHC could be added in text.
2. Lines no. 69 and 70: “generating tolerant genes” can be replaced by other appropriate scientifically appropriate word.
3. Line No. 79: Italicize ‘Brassicaceae’
4. Line No. 325: Citation for the ABA signaling pathway should have been from the original/primary report rather than citation from other referring article.
5.Sub headings under Material and Method and Result section can be kept in bold for better highlighting and differentiation of sections

Experimental design

The experimental design has been well sought of but some of the suggestions and questions pertaining to the current study need to be addressed as mentioned below:
1. Line No. 127: Briefly elaborate on significance of ratio to base errors Q20, Q30, and GC in context of present study and how they are calculated.
2.Line No. 134: Composition of the lysis buffer should be provided in the supplementary material or the reference for the same should be cited in the text.
3.Several technical points needs to be addressed by the authors pertaining to transcript and proteome analysis:
a) Comparative total number of read in control (non-stress) and stress (drought) post transcript analysis has not been mentioned in the result section.
b) What was the average read length obtained and corresponding read length with respect to sequence obtained can be provided as tabular form as supplementary table.
c) Average read length of DGs after transcript assembly particularly unigenes can be mentioned briefly in result section. Likewise for proteome analysis, total number of unique proteins identified and total number of detected peptide spectra can be mentioned for better understanding of the readers.
4. Sample preparation for L2-L1 and L3-1 transcript analysis seems to be slightly unclear particularly to whether the RNA samples for specific treatment were pooled in the given order or the pooling of samples were done only for the biological replicates? Kindly elaborate on the same for sample preparation carried out for proteomic analysis.
5. Line No. 161: Apart from β Actin gene, was the normalization of q-RT data performed using any other internal control/House-keeping genes? If not then how was the differentiation in expression of housekeeping genes assessed and normalized?

Validity of the findings

1. The authors have extensively carried out functional annotation using gene ontology and other tools for functional classification of identified differential genes and proteins. Although the data is provided in the supplementary material, it could be better if a proportionate-wise graphical representation (apart from bar diagram) summarizing genes and protein (separately) as per functional characterization can be provided in main text.
2. Quality of images and figures needs to be improvised. Axis title for Fig. 1 (B, C, D, E, F) can be kept in proximity to the axis. Tick marks should be uniformly formatted (either outside or inside).
3. In figure 4 fill of the bar can be changed to have better differentiation between the treatments. Furthermore, error bars and statistical significance should be added in the q-RT expression result.
4. Lines 36-43 of method section only highlights conditions used for growing seedling. It would be better to briefly include methodology involved in the current study.
5. The overall findings and the essence of present study needs to be specifically mentioned in the discussion section. The authors have specifically mentioned role of antioxidant enzymes and HSPs in imparting drought tolerance. Following which the discussion part highlights several regulatory pathways and genes to be involved in abiotic stress tolerance making discussion more a generalized statement rather highlighting the salient findings of the current research. Therefore its is suggested that author may re-write the discussion and conclusion section making the objective and importance of current study more clear to the international readers.

Reviewer 3 ·

Basic reporting

While the study is quite important in the field of plant stress biology, the quality of writing of the manuscript needs to be improved. For example, the use of the words accelerated (line 37; adjusted (Line 51 and throughout the manuscript) are not appropriate for use. Furthermore, in the abstract and elsewhere in the manuscript, all abbreviations should be defined at first mention. There are numerous errors in sentence construction, which I am unable to list all down. I would recommend that the authors have their manuscript read by colleague who is proficient in English or even professional language editors – otherwise the expensive work conducted in the study becomes difficult to comprehend.

Furthermore, the materials and methods section needs to be extensively revisited. For example, lines 106-107 are not written in the appropriate manner. In line 112, it is not clear what the experimental design was like. Lines 117-120 are not clear if the physiological measurements were only done in the control samples or what? During cDNA synthesis, it is not clear how the authors determined the “high quality of RNA”? Components of the lysis buffer (line 134) need to be specified or a reference given. It is also not clear to me what type of RNA isolation kit was used and whom the manufacturer is (Line 124). It is not clear to me what criterion was used to select the genes for qPCR validation (lines 214 etc). These are just a few of the comments I have. The entire methods section needs extensive revision, and so does the entire manuscript.

Experimental design

The authors conducted a transcriptome and proteome analysis in their plant of interest. However, justification of why they used these two high-throughput methods should be given. Furthermore, it is not clear to me why the different water stress regimes were used, why MDA content was measured etc. A lot of justification in the experimental design used should be given as well.

Validity of the findings

It is quite difficult for me to get to the result and comment on validity if the basic reporting style needs a complete revisit.

Additional comments

Unfortunately, the manuscript is not ready for publication in its current state. However, if re-written and re--submitted I am willing to review the manuscript once again. The work is quite important in the field of plant stress biology, but not in the current state.

---

## Round 0.2 · Minor Revisions

Thanks for the updates. The manuscript is in better form now. But it is not ready for publication yet - the reviewers still have some critical comments. Please consider the detailed remarks by reviewer #3. Please check grammar and misspellings throughout the text as well.

·

Basic reporting

The authors have done modifications in the relevant sections as suggested. All the queries have been addressed appropriately.

Experimental design

“weighing method” should be replaced by gravimetric method in line No. 37.

Validity of the findings

'no comment'

Additional comments

Conclusion should give major outcome of the study in the light of future perspective to have insights into drought tolerance mechanism. Therefore it could be made more concise.

·

Basic reporting

In the manuscript entitled “Understanding the response in Pugionium cornutum (L.) Gaertn. seedling leaves under drought stress using transcriptome and proteome integrated analysis” the authors have reconstructed the material and method and discussion section as suggested concisely covering all the key aspects of the current work. All the queries and suggestions have been answered and incorporated appropriately. However, certain suggestions for further improvement and better understanding of international audiences have been mentioned as follows:
1. Line No. 42: “18:00”, hour or ‘Hr’ should be added.
2. Several grammatical and sentence framing errors throughout the manuscript can be checked. For eg. “Line No.98 “that resisted environmental stress” should be replaced with “that resists environmental stresses”.
3. In line No. 851: hyphen should be added between ABA and PYL/RCAR to signify ABA-PYL complex formation.
4. In line No. 582: Sucrose non-fermenting receptor kinase-2 (SnRK2) is written as SnRK2J. Does it signify any other kinases or is it a typing error?
5. The conclusion seems to be lengthy and more of reiteration of result and discussion section. It is suggested to author to re-frame conclusion section wherein it gives essence or the outcome of the present work and how it will add on to knowledge enhancement for future research
6. Spacing between the sentence and words throughout the manuscript needs to be edited.
7. The authors have mentioned full form of “Figure” in the title of figures while in the text abbreviation “Fig” has been used. The authors are suggested to maintain uniformity as per the journals guidelines.
8. The axis title in Figure 1-E and F can be centrally aligned to axis.
9. Font size in Fig. 6 should be same for all the axis titles.
10. Pixel quality of some images needs to be enhanced. For eg.: Figure No. 4, 5 and 7 .

Experimental design

All the key points pertaining to methodology have been well sought and covered in the revised version.

Validity of the findings

No Comment

Additional comments

No comment

Reviewer 3 ·

Basic reporting

The quality of basic reporting has greatly improved, although more still needs to be done in terms of improving clarity in writing. My comments are as follows:


ABSTRACT
1. I found it difficult to under the meaning of L1, L2 and L3 (Lines 38-39). What does the L stand for? If it is the drought stress levels, then I suggest that the authors include (L) in Line 38 after the word “levels”. In the same note, I also struggled to understand what L2-L1, L3-L3 Lines 46-51 mean. Please clarify this.
2. It is also important to clearly state what the well-watered control treatments was in the study as this is not specified in the abstract.
3. Line 42: The conditions under which the “seedlings were increasingly wilted” need to be clearly stated in that line.
4. The sentence in Line45-50 has too much information, is too long and difficult to understand. Please rewrite the information in 2 shorter sentences.
5. Line 49: The word “distinguished” seems inappropriate. Please revisit.
6. Line 50: “The integrated analysis” needs to be clarified.
7. Some abbreviations are defined in the abstract yet others are not. Please consult the journal’s instructions to authors for guidelines. I suppose there should be some form of consistency in how this aspect is handled.
8. The Methods section in the abstract seems incomplete as it only described the plant growth methods and not the physiological, biochemical and molecular analyses. Please consult the journal’s instructions to authors for guidelines.


INTRODUCTION
The introduction is appropriate for the study and generally well-written. I thank the authors for taking my previous comments seriously. Below are few comments.
1. Line 65: I am not sure how a decrease of cell turgor helps plants to survive drought stress. please check on this.
2. Line 68: Replace “environment stress” with environmental conditions” or “cases”.
3. The authors should just check that they do not place a period before a ref citation e.g. in Lines 72, 86 and so on.
4. Line 76 needs revision. What is Brassicaceceae Pugionium? the plant is described as a biennial herb in Line 76, but an annual or biennial plant in Line 33. So, which is correct?
5. The location of the places cited in Line 77 -78 should be stated. Is this in China or where?
6. Line 80: replace “the” with “a”
7. Line 83; What does “sand fixation” mean?
8. Lines 88-89: Delete “the effects of”
9. Lines 94-98. I suggest that the authors restructure the sentence as follows: “Thus, the …………changes in genes and proteins expression profiles of P. cornutum leaves under drought stress, and understand the regulatory pathways of the plant in drought response.
10. In line 98, I suggest the authors revise the phrase “sequence information”. Maybe use terms line “molecular data” or “genetic information” etc.



MATERIALS AND METHODS

The quality of writing of the materials and methods section need to be extensively improved. I suggest that the authors also check on how other published manuscripts in their field of research are written. Due to the enormity of text to be corrected, I am unable to list all of them. However, below are some of my comments.

1. Line 103: write as ………China, which do not need specific permits were collected (Wang et al. 2017).
2. In Line 104, it is not clear how the seeds were “cultured in an incubator”. What was the growth medium, temperature settings etc? was this under plant tissue cultures conditions?
3. The structure of text in Lines 104-107 needs to be improved.
4. Line 108. After reading the methods, I am still unsure what the weighing method is and what it involved. This needs to be clarified. Furthermore, the authors need to cite a reference citation for this weighing method in Line 108.
5. The words “consistent” in Line 109, “ripe” in Line 115, “know” in Lines 119 & 177, “detected” in Line 130, “detect” in Line 162, seem inappropriate. Please revise.
6. In Line 117, did the authors mix leaf and roots samples together? This needs to be clarified.
7. Line 122: delete “The determination methods”.
8. Line 134: The authors could not have conducted protein sequencing using total RNA extracts. This is contextually incorrect and should be correct.
9. Line 136: replace “has” with “with”
10. Line 138: Please correct sentence construction as you cannot begin a sentence with “And”.
11. Line 144: what is SDT?
12. All rpm values should be converted to x g values. Please consult the manuals of the centrifuges used for these conversions. However most current micro-centrifuges have an automatic conversion knob on them.
13. There are places were reference citations have the authors initials included. Please correct this e.g. in lines 148, 153 etc.
14. It is not possible to have digested protein from leaves (Line 151). Please correct this, and where possible avoid beginning sentences with a numerical number (Lines 151, 163, 191 etc).
15. Line 153. Please correct sentence as you could not have analysed MS data with a mass spectrometer.
16. Line 157: Is it DGs or DPs?
17. Line 158-160 needs structural changes.
18. What does “drought stimulation” mean?
19. Which transcriptome database was used? Line 165
20. The information on the kit used for cDNA needs corrections. Please check in our published papers how this is correctly done. The same goes for the “qPCR procedure of the TAKARA company” text in Line 169.
21. It is not clear what data was analysed and how it was analysed (Lines 172-173). This needs revision.




RESULTS
The results section is well-written for such an extensive multi-omics study. Below are a few comments.
1. The sentence in this section needs revision.
2. In Line 180, what is WHC? Did the authors mean RWC?
3. What does it mean to say the leaf water potential increases (181-182)?
4. Line 183-184 needs revision.
5. Line 184: write as: The plant physiology results were consistent with …..
6. The word positively in Line 185 is inappropriate.
7. Line 186-187: Next, P. cornutum leaves were harvested for use in transcriptome and proteome analyses under drought stress.
8. Where possible, do not discuss results in the results section. Rather save that for the discussion.
9. Line 203-204 needs revision.
10. You cannot begin a sentence with “Of which”. Rather use “Of these” as a follow-on sentence to the previous one.
11. Line 207; write as … elaborating the functions of these DGs.
12. Line 220. In the back of except…… needs revision.
13. What is a multiple expression difference?
14. Figures 4 and 5 are not legible.
15. Line 239-240. Write as “To validate the proteomic data, seven randomly DPs were selected for qPCR analysis.
16. In Line 241: what is thermal stimulation? Why did you select this functional grouping for qPCR analysis?
17. Line 241-243 need revision.
18. The numbers in Line 251 are rather confusing to read and comprehend. Please revise.
19. What does jointly mean? Line 253?
20. Line 256. It is worth noting ….
21. Line 266: A small number of genes….
22. The sentence in Line 275-276 seems misplaced.


DISCUSSION
1. Generally, the discussion is appropriate for the study but the authors need to
a. Be cautious that their study was limited to some (not all) physiological experiments, transcriptome and proteome analyses. Therefore, the discussion should not seem as if downstream experiments were done to validate activity of enzymes etc (e.g. in Lines 349-350). Please correct this.
b. There is also a lot of sentence and language issues to be corrected in this discussion and I suggest this is done.
c. There are also places where the authors are re-writing text in the results section as it is. This should be corrected. While the trends in results ought to be re-stated in the discussion, the authors are to discuss their results.
2. Line 291-292: please correct the context of this sentence.
3. Line 295: maybe add the word “events” after molecular
4. Line 296. Replace the word “In” with “Under”
5. Line 297. It is no clear what happened with the “higher proline content”
6. In the discussion, please avoid texts line “the above-mentioned studies” because it just reduced clarity of facts.
7. Texts in Lines 303-308 need adequate referencing to give credit to Shinozaki and others who are pioneers in this work.
8. Please revise Line 310-312.
9. You cannot begin a sentence with “whereas”. Line 315.
10. Line 322-323: please list the relevant refs here to support your statement.
11. Line 324: please correct ref citation in this sentence.
12. Lines 324-351: need extensive revision and language editing.
13. Please ensure that you correctly cite references, not only in the names of authors, but in the type of study they did as well. For example, the ref Tatenda et al., 2020 is actually Goche et al., 2020. Furthermore, that study by Goche et al., 2020 looked at drought stress responses in sorghum and NOT salt stress. Please do check that all your ref citations are correct.
14. Line 346-348 seems lost. Either delete it or relate it to your study.
15. What does “notably expressed” mean? Do you want to say “differentially expressed”?
16. In the results and discussion sections, the authors write about “significantly induced and depressed” DGs or DPs, then go on in the same sentence to state which DGs or DPs were up or down regulated. My suggestion is that the authors replace “significantly induced and depressed” with a general term such as “differentially expressed”. In this way, the sentences become easier to understand.
17. In lines 357-360: I suggest that the authors give examples of these interactions and also corrects the sentence structure of Line 359-360.
18. Also correct sentence strudture in Lines 363-367.
19. Avoid phrases like “It is not hard to see” Line 390
20. In Line 392. Replace the word “metabolism” with “metabolic”
21. Leaves (Line 404)
22. Line 405. How did the authors come up with the conclusion on the synergestic effects on stomatal movements (line 405) when the study did not look at stomatal closure etc? Please correct and discuss your study results within the confines of your study.

CONCLUSION
Please re-write the concluding remarks in such a manner that they are not a mere repetition of the results section.

Experimental design

The experimental design is sound, but the materials and methods section needs extensive revision. please see my comments above.

Validity of the findings

The findings of the results are also good

Additional comments

Please see my comments attached under the basic reporting section.

---

## Round 0.3 · Minor Revisions

Thanks for the update and detailed answer. I see that all the reviewers’ remarks were considered in the revised version. However the Section editor added some remarks demanding revision.

You need to update some figures, add accompanying tables. Check again English, and make language proofing.

See the comments below:

Though transcriptome and protein analysis are discussed, the highlighted gene classes exist by name only; there is no concrete way for the reader to validate such claims. There needs to be a pointer to some of the raw data in its processed form so that matches can be made between the classified annotation figures.

The Venn diagrams listed in Figure 2 need accompanying tables listing the identified genes contained within. An accompanying table needs to explained classifications shown in the figures 4 and 5 to match genes identified, GO terms, and annotations.

The manuscript may have some value, but in its current form it is only hearsay without any concrete data provided. I would have also expected some of the peptide vs sequence data to be matched in some fashion where codon usage might be assessed, or at least some anchor to available sequence data as simply selected annotations are quite error-prone. The manuscript may have valuable assessments, but in its current state it is lacking vital information to provide value to the reader. Revisions are suggested. There are some areas which needed some language proofing.

---

## Round 0.4 · accepted · Accept

Thanks for the update. All the minor remarks were considered. We shall accept this work for publication in current form.

The Section Editor suggests that you provide your data ain a more accessible format such as ZIP.